# Impact of Cooking Methods on Phenolic Acid Composition, Antioxidant Activity, and Starch Digestibility of Chinese Triticale Porridges: A Comparative Study between Atmospheric Pressure and High Pressure Boiling

**DOI:** 10.3390/foods13020230

**Published:** 2024-01-11

**Authors:** Hua Li, Yurong Mao, Danni Ma, Hua Li, Ruixin Liu, Sirithon Siriamornpun

**Affiliations:** 1Department of Cuisine and Nutrition, Yangzhou University, Yangzhou 225127, China; 2Research Unit of Thai Food Innovation (TFI), Mahasarakham University, Kantarawichai 44150, Thailand; 3Department of Food Technology and Nutrition, Faculty of Technology, Mahasarakham University, Kantarawichai 44150, Thailand

**Keywords:** phenolic acid, DPPH radical scavenging ability, ABTS radical scavenging ability, resistant starch, starch digestion, glycemic index, cereal

## Abstract

Water boiling under atmospheric pressure (CAP) and water boiling under high pressure (CHP) are two popular domestic cooking methods for Chinese porridge making. In this study, we aimed to evaluate the effects of these two methods on the phenolic acid composition, antioxidant activity, and starch digestibility of triticale porridges. The contents of total free and total bound phenolic acids in the CHP sample were 1.3 and 1.6 times higher than those in the CAP counterpart, respectively, although the DPPH and ABTS values of these two samples were comparable. CAP induced more small pieces of starch than CHP, and the gelatinization enthalpy was 19% higher in the CHP sample than that in the CAP. Both cooking methods increased the starch digestibility, while the CHP sample (58.84) showed a lower GI than the CAP (61.52). These results may promote the application of triticale in health-promoting staple foods.

## 1. Introduction

Triticale (× *Triticosecale Wittmack*), crossed by wheat (*Triticum* sp.) and rye (*Secale cereale* L.), shows some excellent agronomic traits such as drought resistance and low susceptibility to pests and diseases [1], and it is also abundant in starch, protein, dietary fiber, and minerals, as well as bioactive compounds including phenolic acids, anthocyanins, and pentosan [2]. Previous studies have shown that triticale has in vitro antioxidant activities [3] and could improve the metabolism of blood lipids in hyperlipidemia rats [4], as well as glucose tolerance in diabetic mice [5]. Triticale has always mostly been used as animal feed, but with a deepening understanding of its nutritional value and health-promoting effect, it has been increasingly applied to the production of a wide range of foods, such as bread, steamed bread, noodles, biscuits, and cakes [2].

Among the bioactive compounds in cereals, phenolics are the most diverse and complex. In addition to their antioxidant, anticancer, and anticardiovascular disease effects, phenolics can delay the conversion of starch to glucose by inhibiting α-amylase and α-glucosidase, thus playing a positive role in the control of postprandial blood glucose in diabetics [6]. Phenolic acid, the most common form of phenolics in cereals [7], plays a significant part in protecting the human body from oxidative damage. There were significant differences in phenolic acid composition and antioxidant capacity among the varieties of wheat, rice, millet, and other cereals [8,9,10]. Meanwhile, food processing may improve or decrease the phenolic level and antioxidant capacity of the cereals [11]. Moreover, the effects of thermal processing techniques, including boiling, steaming, roasting, baking, and extrusion, on phenolic level and antioxidant activity varied with the temperature and time of these heat treatments and varieties of cereals [7,12]. Although much research has been done on the effects of different kinds of processing methods on the levels of phenolic compounds and antioxidant activity in some cereals, little work has been done on the impacts of domestic cooking on the phenolic acid composition and antioxidant capacity of triticale.

Starch, as the most predominant nutrient in triticale, can be categorized into three types, i.e., rapidly digestible starch (RDS), slowly digestible starch (SDS), and resistant starch (RS), according to its digestibility. If starch is digested quickly in the human body, it may lead to a sharp rise in postprandial blood sugar, thus causing some metabolic diseases, such as insulin resistance and diabetes [13]. Previous studies have shown that heat processing has different effects on starch digestion. For example, Rehman and Shah [14] found that boiling at 121 °C for 10 min could improve the digestibility of bean starch, but Pan et al. [15] found that brown rice starch did not completely gelatinize after cooking for 60 min, showing strong anti-digestibility. Moreover, cooking may have little effect on starch digestion. Wu et al. [16] found that after steaming, the RDS and SDS contents of rice only increased by 4.5% and decreased by 5.1%, respectively. To the best of our knowledge, the starch digestibility of boiled triticale has not been reported on.

China stands as a prominent producer of triticale within the Asian region, yielding an impressive 449,306 tons in 2019 [17]. Porridge holds a distinct position among popular staple foods in China, with cereals being a prevalent choice for its preparation within households. Presently, the two most widely employed methods for porridge preparation are water boiling under atmospheric pressure (CAP) and water boiling under high pressure (CHP). However, no information has been released on how these two methods affect the nutritional quality and other health-promoting properties of triticale porridge. Our study focused on elucidating the phenolic content, phenolic acid composition, antioxidant capacity, and starch digestibility of a prevalent triticale cultivar in northern China. Furthermore, we aimed to assess the influence of the aforementioned cooking methods on these characteristics of triticale. The experimental designs are summarized as shown in Figure 1. This investigation serves to enhance our comprehension of the impact of domestic cooking on the bioactive compounds and starch digestibility of triticale, thereby providing valuable insights for optimizing triticale cooking practices and developing a diverse range of triticale-based food products.

## 2. Materials and Methods

### 2.1. Triticale and Chemicals

The triticale cultivar ND2201 (2 kg) was supplied by Huinong Fumin Technology Co., Ltd. (Beijing, China). Folin–Ciocalteu reagent, 2,2-diphenyl-1-picrylhydrazyl (DPPH), 2,2′-azino-bis(3-ethylbenzothiazoline-6-sulfonic acid) (ABTS), and 6-hydroxy-2,5,7,8-tetramethylchroman-2-carboxylic acid (Trolox) were obtained from Sigma-Aldrich Fine Chemicals (St. Louis, MO, USA), and alkaline protease, pepsin, and standards of ferulic acid (FA), protocatechuic acid (PRCA), *p*-hydroxybenzoic acid (*p*-OHBA), vanillic acid (VA), *trans*-Cinnamic acid (*t*-CA), *p*-coumaric acid (*p*-CA), gallic acid (GA), syringic acid (SRA), caffeic acid (CA), sinapic acid (SA), and paracetamol (internal standard, IS) were acquired from Yuanye Bio-technology Co., Ltd. (Shanghai, China). Glucoamylase and α-amylase were attained from Aladdin Scientific Corp. (Shanghai, China), and HPLC-grade methanol and acetic acid were purchased from J&K Scientific Ltd. (Beijing, China). All other solvents purchased from Sinopharm Chemical Reagent Co., Ltd. (Shanghai, China) were of analytical grade.

### 2.2. Preparation for Raw and Cooked Triticale Samples

The triticale grain (350 g) was ground, and then the powder passing through a 0.5 mm sieve was stored at −20 °C until use.

A triticale porridge was prepared by CAP as follows. After being rinsed with water, 40 g of triticale grain, together with 320 g of water at 50 °C, was put in a rice cooker (CFXB50HC2-120, Supor, Shaoxing, China). After being kept warm for 30 min, the triticale was cooked for 90 min by using the panel function “porridge”, and then the cooked mixture was freeze-dried, followed by grinding and passing through a 0.5 mm sieve. Subsequently, the powder was stored at −20 °C until use.

Similarly, a triticale porridge was prepared by CHP. In total, 40 g of triticale grain, together with 200 g of water at 50 °C, was put in an electric pressure cooker (SY-50HC18Q, Supor, Shaoxing, China). After being kept warm for 30 min, the triticale was cooked for 60 min at 60 KPa using the panel function “coarse cereals porridge”.

### 2.3. Preparation of Triticale Flour without Lipids or/and Proteins

The lipids and proteins were removed from the raw and CHP triticale flours according to the method of Ye et al. [18]. Briefly, 100 g of triticale flour was delipidized with 500 mL of petroleum ether at room temperature, followed by filtration to obtain the solid residue. The above process was repeated twice to obtain triticale flour without lipids. For protein removal, 100 g of triticale flour was mixed with 800 mL of carbonate buffer (0.02 M, pH 9.0) containing alkaline protease (120 U/mL) and hydrolyzed at 45 °C for 1 h in a water bath. After centrifugation at 4000× *g* for 10 min, the precipitate was collected, and the hydrolysis was performed twice using the same procedure. Subsequently, the obtained residue was repeatedly washed to neutrality with distilled water. Triticale flour without lipids and proteins was prepared by sequentially removing lipids and proteins as described above.

### 2.4. Extraction of Free Phenolics

Free phenolics in the raw and cooked triticale samples were extracted according to Tian et al. [19] with a little modification. In total, 1 g of triticale powder was mixed with 20 mL of 70% methanol and sonicated for 15 min at 40 °C with an ultrasonic extractor (XH-2008D, Xianghu Technologies, Beijing, China), followed by centrifugation (Allegra X-22R, Beckman Coulter, Brea, CA, USA) at 3000× *g* for 10 min at 4 °C. The residue was extracted again following the same method, and the combined supernatants were evaporated to less than 2 mL under nitrogen at 30 °C. Afterwards, the concentrate was mixed well with 2 mL of 1% acetic acid and injected into an SPE cartridge (Oasis HLB, 6 cc, 200 mg; Waters, Milford, MA, USA). To remove sugars and other polar constituents, the cartridge was washed with 3 mL of water. Subsequently, 2 mL of methanol/1% acetic acid (9:1, *v*/*v*) was used to elute the absorbed compounds. Finally, after the eluates were evaporated to dryness under nitrogen, the residues were redissolved in 200 μL of methanol/water (1:9, *v*/*v*) containing the IS (paracetamol, 10 μg/mL) and then passed through a 0.45 µm membrane (Jinteng, Tianjing, China).

### 2.5. Extraction of Bound Phenolics

Bound phenolics in the raw and cooked triticale samples were extracted according to Zieliński et al. [20] with a slight alteration. The mixture of the precipitate from the extraction of free phenolics and 20 mL of 4 M NaOH was sonicated for 90 min at 40 °C, and then the pH of the solution was adjusted to 2.0 using concentrated HCl, followed by centrifugation (Allegra X-22R, Beckman Coulter, Brea, CA, USA) at 3000× *g* for 20 min at 4 °C. Subsequently, the released phenolics were extracted three times with 30 mL of ethyl acetate from the supernatant. The combined ethyl acetate extracts were dried using a rotary evaporator (RE-52AA, Yarong, Shanghai, China), and then the dry residue was dissolved in 1 mL of methanol/water (1:9, *v*/*v*) containing the IS (paracetamol, 10 μg/mL) and passed through a 0.45 µm membrane (Jinteng, Tianjing, China).

### 2.6. Determination of Total Phenolic Content

The total phenolic contents of free and bound extracts from the raw and cooked triticale samples were measured with three repetitions using the Folin–Ciocalteu method reported by Kubola et al. [21]. A standard calibration curve was prepared using gallic acid (GA; 5, 15, 25, 35, and 45 μg/mL), and the total phenolic contents of the extracts were calculated based on the standard curve and expressed as milligrams of gallic acid equivalent per gram of dry sample (mg GAE/g).

### 2.7. Analysis of Phenolic Acid Composition

The contents of each phenolic acid in free and bound phenolic extracts from the raw and cooked triticale samples were evaluated via three repetitions using the HPLC method described by Irakli et al. [22] with minor adjustments. A 20 μL aliquot of sample solution was fractionated using an HPLC system (LC1200, Agilent Technologies, Santa Clara, CA, USA) with a diode array detector. An Inspire C18 column (5 µm, 4.6 mm × 250 mm; Dikma tech, Lake Forest, CA, USA) was used for fractionation at 30 °C. The mobile phase, with 1 mL/min of the flow rate, consisted of solvent A (methanol) and solvent B (acetic acid/water, 1:99, *v*/*v*). A 58 min linear gradient was performed as follows: 0–13 min, 90–80% B; 13–15 min, 80–75% B; 15–28 min, 75–65% B; 28–43 min, 65–35% B; 43–52 min, 35–0% B; 52–58 min, 0% B. The detected wavelengths were 260 nm for IS, PRCA, *p*-OHBA, VA, and *t*-CA, 270 nm for GA and SRA, and 320 nm for CA, *p*-CA, FA, and SA. The spectra were recorded from 190 to 400 nm. Phenolic acids in the free and bound phenolic extracts were recognized by comparing their UV spectra and relative retention times with those of the authentic compounds, and the concentration of each compound was calculated based on the peak area ratio of analytes to IS at 10 μg/mL using an internal standard method. The phenolic acid content was expressed as micrograms per gram of dry sample (μg/g).

### 2.8. Determination of DPPH Radical Scavenging Activity

The DPPH radical scavenging activities of free and bound phenolic extracts from the raw and cooked triticale samples were measured with three repetitions using the method of Mareček et al. [23]. The radical scavenging activity was calculated on the basis of the Trolox standard curve and expressed as micromoles of Trolox equivalent per gram of dry sample (μmol TE/g).

### 2.9. Determination of ABTS Radical Scavenging Activity

The ABTS radical scavenging activities of free and bound phenolic extracts from the raw and cooked triticale samples were evaluated with three repetitions according to the method provided by Mareček et al. [23]. The radical scavenging activity was calculated based on the Trolox standard curve and expressed as micromoles of Trolox equivalent per gram of dry sample (μmol TE/g).

### 2.10. Determination of Ferric Reducing Antioxidant Power (FRAP)

The abilities of free and bound phenolic extracts from the raw and cooked triticale samples to reduce iron (III) were assessed across three repetitions using the method of Senol et al. [3]. The FRAP was expressed as micromoles of Trolox equivalent per gram of dry sample (μmol TE/g).

### 2.11. Analysis of Starch Digestibility

#### 2.11.1. Determination of Rapidly Digestible, Slowly Digestible, and Resistant Starch Contents

The levels of RDS, SDS, and RS in triticale samples were evaluated across three repetitions following the method of Englyst et al. [24] with some modifications. Briefly, 15 mL of sodium acetate buffer (0.2 M, pH 5.2) was vortexed with triticale flour (600 mg) and guar gum (50 mg). After adding 5 mL of enzyme solution (porcine pancreatic α-amylase, 1200 U/mL; glucoamylase, 800 U/mL), the mixture was incubated at 37 °C and 160 rpm in a water bath. After reaction for 20 and 120 min, 1 mL of the supernatant was taken, and 5 mL of pure ethanol was immediately added to deactivate the enzyme. Then, the enzymatic hydrolysate was centrifuged at 2500 rpm for 25 min, and the glucose content in the supernatant was determined by the DNS method [6].

#### 2.11.2. Determination of In Vitro Glycemic Index

The in vitro glycemic index (GI) was measured with three repetitions based on the method of Goñi et al. [25] with a little modification. Briefly, 500 mg of triticale flour was vortexed with 20 mL of phosphate buffer (pH 7.5), and 1 M hydrochloric acid was added to bring the pH down to 1.5. Following the addition of 0.4 mL of pepsin solution (115 U/mL), the mixture was incubated for 30 min at 37 °C in a water bath and then immediately cooled to room temperature. Subsequently, the enzymatic hydrolysate was adjusted to pH 6.9 using 1 M sodium hydroxide and diluted with phosphate buffer (pH 6.9) to 50 mL. After taking 1 mL of the solution as the sample at 0 min, 0.1 mL of α-amylase solution (40,000 U/mL) was added to the remaining solution and shaken in a water bath at 37 °C. Then, 1 mL of enzymatic hydrolysate was taken at 30, 60, 90, 120, 150, and 180 min and immediately deactivated in a boiling water bath for 5 min. After being cooled to room temperature, 3 mL of sodium acetate buffer (0.4 M, pH 4.75) and 0.1 mL of glucoamylase (3300 U/mL) were added and then shaken for 45 min at 55 °C in a water bath, followed by deactivating the enzyme. The glucose content of the enzymatic hydrolysate was determined by the DNS method [6], and the GI of triticale was estimated according to the formula below:GI = 39.7 + 0.549 × HI

### 2.12. Scanning Electron Microscopy (SEM) Analysis

The morphological characteristics of raw and cooked triticale flours were observed by scanning electron microscopy (GeminiSEM300, Carl Zeiss, Jena, Germany). The freeze-dried flours were mounted on the sample plate and then coated with gold to make the sample conduct electricity. The images were photographed with a magnification of 1000× at an accelerating voltage of 5 kV.

### 2.13. X-ray Diffraction Analysis

The X-ray diffraction patterns of raw and cooked triticale flours were investigated across three repetitions using an X-ray diffractometer (D8 Advance, Bruker AXS, Karlsruhe, Germany) at 30 kV and 40 mA with Cu-Kα radiation. The scanning region of diffraction angle (2θ) ranged from 5° to 40° at a scanning speed of 2°/min. The relative crystallinity was calculated using the X-ray diffractometer software (Topas 4.0).

### 2.14. Thermal Characteristic Analysis

The thermal properties of raw and cooked triticale flours were studied across three repetitions using a differential scanning calorimeter (DSC-8500, PerkinElmer, Waltham, MA, USA). Briefly, 3 mg of each flour was weighed into an aluminum crucible, and then distilled water was added at a ratio of 1:2 (flour/water, *m*/*v*). After being sealed, the crucible was kept at 4 °C for 12 h, followed by heating from 25 °C to 100 °C at a constant rate of 10 °C/min. The onset temperature (*T*o), peak temperature (*T*p), conclusion temperature (*T*c), and gelatinization enthalpy (Δ*H*) were attained using the DSC software (Pyris 11).

### 2.15. Statistical Analysis

The results have been expressed as mean ± standard deviation (SD) for three replications. The data were evaluated by a one-way analysis of variance (ANOVA) test followed by Duncan’s multiple range test in SPSS statistical software (version 16.0; Chicago, IL, USA). The level of significance was set at *p* < 0.05.

## 3. Results and Discussion

### 3.1. Free, Bound, and Total Phenolic Contents

The phenolic contents of raw and cooked triticale samples are displayed in Figure 2. For the raw sample, the free and bound phenolic contents were 0.67 and 0.53 mg GAE/g, respectively, lower than the levels reported by Gan et al. [26]. This may be attributed to the difference in the cultivar of triticale and the extraction method of phenolics.

Boiling, as one of the most basic methods of starch gelatinization, is widely used in household grain processing. In this study, we prepared triticale porridge of acceptable organoleptic quality using two ways of cooking: long-time boiling at atmospheric pressure and short-time boiling at high pressure. After cooking, no significant change was observed in the free phenolic content, which is similar to the findings of Scaglioni et al. [27] that no significant change was noticed in the parboiled rice after boiling. Nevertheless, Scaglioni et al. [27] also found that the content of free phenolic compounds in the polished and whole rice significantly decreased after boiling. In contrast, Bryngelsson et al. [28] indicated that cooking could effectively increase the free phenolic content of oats. The discrepancy may be attributed to differences in the binding, release, degradation, and depolymerization of phenolics during thermal processing [11].

After cooking, the content of bound phenolics significantly increased by 64% (CAP) and 79% (CHP). This may be partly related to the destruction of the cell wall structure and the partial hydrolysis of fibrous polysaccharides due to high cooking temperatures [29]. In addition, Guo [30] found that with the increase in pressure, the total phenolic content in triticale bran increased continuously and reached its highest value (6.40 mg/g) at 1.0 MPa—2.6 times higher than that of its untreated counterpart. Thus, high pressure may also promote the release of phenolics, which is also a key factor influencing the content of total phenolics in high-pressure-cooked triticale.

Besides boiling, there are many other methods of grain thermal processing, such as steaming, baking, and microwaving. The phenolic content before and after processing often varies with grain types and thermal processing methods. For example, Pradeep and Sreerama [10] found that microwaving and steaming induced an evident increase in the total phenolic content of millets. On the contrary, some thermal processing treatments, i.e., autoclaving, microwaving, and baking, significantly reduced the total phenolic content of buckwheat flour [31]. Furthermore, no change was observed in the total phenolic contents of dark and white buckwheat flours after roasting for 10 min at 200 °C [32]. These results indicate that due to the difference in the temperature, pressure, and duration of various thermal processes, even for the same grain varieties, the phenolic content may vary. Therefore, to broaden the application of triticale in food products, further studies may be conducted on the influence of other food processing methods on the bioactive compounds, such as phenolics, in triticale.

### 3.2. Free and Bound Phenolic Acid Compositions

Phenolic acids can be divided into two groups: hydroxybenzoic acid derivatives, such as gallic, *p*-hydroxybenzoic, syringic, protocatechuic, and vanillic acids, and hydroxycinnamic acid derivatives, such as caffeic, cinnamic, ferulic, *p*-coumaric, and sinapic acids [33]. The phenolic acids in cereals are mainly present in their bound form, and are linked to structural components of the cell wall such as cellulose, lignin, and proteins by ester bonds [34]. In this study, we analyzed the concentrations of ten common phenolic acids in the raw and cooked triticale extracts using the HPLC method, and the results are presented in Table 1. To get more precise results, we may use HPLC-MS in our future studies.

For the raw sample, the contents of total free and total bound phenolic acids were 42.96 and 868.11 μg/g, respectively. *t*-cinnamic acid (18.16 μg/g) was the most abundant free phenolic acid, followed by ferulic (6.85 μg/g), vanillic (5.02 μg/g), and *p*-coumaric (3.75 μg/g) acids, while ferulic acid (782.49 μg/g) was the most abundant bound phenolic acid, making up more than 90% of the total bound phenolic acids, followed by sinapic (27.54 μg/g), vanillic (19.41 μg/g), and *p*-coumaric (15.94 μg/g) acids. Similar findings were reported by Weidner et al. [35], who also verified that ferulic, *p*-coumaric, and sinapic acids were the predominant phenolic acids in triticale.

Both CAP and CHP brought about a significant decrease in the total bound phenolic acid content, but CHP induced a significant increase in the total free phenolic acid content. Moreover, as for individual phenolic acid, the content of free ferulic acid increased by 53% and 42% for CAP and CHP samples, respectively, which is consistent with the results of Bryngelsson et al. [28], who found that cooking greatly increased the content of free ferulic acid in oats. However, the content of bound ferulic acid decreased by 68% and 48% for the CAP and CHP samples, respectively.

There are two sides to the influence of cooking on the composition of phenolic acids. Heat treatment may lead to the oxidative decomposition of unstable phenolic acids, or make phenolic acids enter the endosperm and create combinations with macromolecules like proteins, thus reducing their extractable properties [28]. Meanwhile, heat treatment may break the chemical bonds between phenolic acids and polysaccharides, or release the bound phenolics in grains by destroying the cell wall structure [20]. Therefore, the type and content of phenolic acids in triticale samples after boiling treatment may depend on which of the above effects is more dominant.

The phenolic compounds in whole grains mainly exist in an insoluble form. As reported, 62%, 75%, and 85% of phenolics were bound in rice, wheat, and corn, respectively [33], and the bound phenolics in barley made up 55–89% of total phenolics [36]. In our study, like other grains, the bound phenolic acid in the triticale samples was predominant, at more than 89%. Meanwhile, ferulic acid, the most abundant phenolic acid, accounted for over 96% of total bound phenolic acids. However, owing to the bran matrix seriously impeding the action of esterase and xylanase, bound phenolic acids are not easily released in the human stomach and small intestine, which explains their low bioavailability [37]. For example, only 2.6% of the ferulic acid in wheat fiber was released in the stomach and small intestine, whereas 95% was released in the colon of the human body [37]. Although bound phenolics have a protective effect against colon cancer [37], free phenolics are more rapidly absorbed in the stomach and small intestine, and go throughout the body to perform other health-promoting functions, such as inhibiting the oxidation of LDL and liposomes [38]. Therefore, to meet the specific health-promoting needs of consumers, the processing methods for triticale foods should be optimized to increase the contents of free phenolic acids or maintain the level of bound phenolic acids. Furthermore, there is a need to evaluate the bioavailability of the phenolic compounds in processed triticale foods after digestion.

### 3.3. Antioxidant Activity

Epidemiological studies have substantiated that the long-term intake of whole grain foods can lower the risk of various chronic diseases, mainly due to the phytochemicals that are abundant in whole grains, such as phenolics, β-glucans, carotenoids, inulin, and lignans [39], and partly attributed to the antioxidant effects of these active compounds, because excessive free radicals can cause oxidative damage to biological macromolecules, leading to an increased risk of chronic diseases [40]. Therefore, antioxidant capacity is an important basis for the health-promoting effect of triticale.

Since the antioxidant capacity of plant samples can be affected by a variety of factors, such as the extraction solvent and testing system used [41], three different methods, namely, DPPH and ABTS radical scavenging and FRAP, were adopted in this study to reflect the antioxidant capacities of the free and bound extracts in the triticale cultivar and its cooked samples. The results are shown in Figure 3. Both CAP and CHP significantly increased the DPPH and ABTS radical scavenging capacities compared with the raw sample, especially for the DPPH value of free phenolic extracts from the CAP sample, which was about 1.76 times higher than that from the raw sample. However, a significant decrease in FRAP value after cooking was observed.

During cooking, factors determining the antioxidant capacity of extracts are very complex. Usually, high temperatures and high pressures may cause the degradation and destruction of some antioxidants, and also promote the liberation of bound antioxidants such as phenolic acids, thereby improving the extraction rate of antioxidants. Therefore, the combined effects determine the types and contents of antioxidants, which are related to the antioxidant potential. The observed inconsistency in the antioxidant capacity could be explained by the variation in the antioxidant composition induced by the two cooking methods with different temperatures and pressures.

Some researchers have also studied the impact of heat treatment on the antioxidant capacities of different cultivars of millet. Pradeep and Sreerama [10] studied the impact of steaming and microwave treatment on the antioxidant capacity of three millet cultivars (Barnyard, Proso, and Foxtail). They found that the IC_50_ value for scavenging the DPPH radical of Foxtail decreased by 20% after steaming, whereas microwave treatment induced little change. In addition, the FRAP of Proso but not Barnyard and Foxtail was significantly improved after steaming. Furthermore, Chandrasekara et al. [42] showed that the DPPH radical scavenging ability significantly decreased in three of the seven millet varieties after cooking, while no significant change was found in the remaining four varieties. These results suggest that the interaction between the grain variety and the thermal processing method may have strong impacts on the antioxidant capacities, as described by our results.

### 3.4. SEM Analysis

Scanning electron micrographs of the raw and cooked triticale flours are displayed in Figure 4. The raw starch granules were mostly oval- or disk-shaped, with a few small spherical or irregular particles attached to the surface, which may be proteins or protein fragments from grinding, or minerals and cellulose [43]. This phenomenon is comparable to that seen in the results of Kandil et al. [44].

Cooking clearly altered the integrality of starch granules. Due to water molecules penetrating into the starch granules during cooking, the starch granules showed a corroded surface and were even broken into small pieces with irregular shapes. As expected, the CAP sample showed more small pieces and less aggregation of starch fragments compared to the CHP sample, which may be related to the longer heating time in CAP.

### 3.5. X-ray Diffraction and Relative Crystallinity

Starch particles have an insoluble, semi-crystalline structure, and can be classified into three basic types, namely, A-type (containing short amylopectin), B-type (containing long amylopectin), and C-type (a mixture of A-type and B-type) crystal structures, according to their characteristics and different X-ray diffraction patterns [45]. The X-ray diffraction patterns of the raw and cooked triticale samples are shown in Figure 5. The raw sample appeared to have obvious single diffraction peaks at 15°and 23° (2*θ*) and an unresolved double peak at 17° and 18°, indicating that the raw starch had a typical A-type crystalline structure. Additionally, the weak single peak at 20° indicates the existence of a V-type crystal structure. After cooking, the peak intensity decreased and the peak width obviously increased, and only two unresolved double peaks at 17° and 20° were observed. However, there is no evident difference in the diffractograms between the CAP and CHP samples. The reason for the change in diffractograms after cooking may be that during cooking, the absorption of water by starch granules induced the hydrogen bonds between starch molecules to be replaced by those between water and starch, thereby forming a crystal structure that was easily destroyed.

Starch crystallinity is an important aspect affecting the characteristics of starch. For starch with an A-type crystalline structure, gelatinization temperature generally increases with the increase in crystallinity [46], and the higher the crystallinity, the more difficult it is for amylase to enter the starch granules, and the weaker the digestibility of the starch [47]. Compared with the raw sample, the crystallinity of the CAP and CHP samples was greatly reduced by about 50% (Table 2), which is consistent with the trend seen in the characteristic diffraction peaks.

### 3.6. Thermal Properties

The thermal properties of the raw and cooked triticale samples are presented in Table 2. The thermal transition temperatures, i.e., onset (*T*o), peak (*T*p), and conclusion (*T*c) temperatures, significantly decreased after cooking, while the values for the CAP and CHP samples were comparable. The higher transition temperatures for the raw sample may be attributed to its highly ordered crystalline structure, which limited the swelling and dissolution of starch [48]. Kan et al. [49] also found that heat treatment, especially boiling, can significantly reduce the gelatinization temperature of chestnut starch due to the destruction of a considerable percentage of long-chain amylopectin.

The gelatinization temperature range (*T*c–*T*o) implies the existence of crystallites with varying levels of stability inside the starch crystalline regions [50]. Compared with the raw sample, the cooked samples showed a significant increase in *T*c–*T*o, indicating lower homogeneity in the crystals [50], which may be induced by inhomogeneity in the structural arrangement of amylose and amylopectin inside the starch granules [51].

The gelatinization enthalpy (∆*H*) correlates with the absence of double-helical order and crystallinity in amylopectin [52]. Boiling may result in a decrease in amylopectin content and crystal structure, and a decrease in gelatinization enthalpy [53]. The highest ∆*H* was observed in the raw sample (5.61 J/g), followed by CHP (2.22 J/g) and CAP (1.86 J/g). A disruption in the double-helical configuration led to a lower requirement for energy to unravel and melt the crystalline structure of the starch in cooked samples. Similar results were found in heat–moisture-treated maize, pea, and lentil starches [54].

### 3.7. Starch Digestibility

Figure 6 displays the starch composition and GI of the raw and cooked triticales. After cooking, the RDS content significantly increased by 90% and 55% for the CAP and CHP samples, respectively. The increase in RDS content may be due to the expansion, fracture or decrease in the crystallinity of starch granules during cooking, which made starch gelatinization easier to recognize for enzymes [55]. The contents of SDS and RS decreased in relation to, or were comparable to, those before cooking, which agrees with the findings of He et al. [56]. Studies have indicated that increasing SDS and RS content often corresponds to a decreased digestibility of starch [57]. As indicated in our results, the CAP sample had the highest GI, while the raw sample had the lowest GI. Moreover, proteins and lipids may also affect starch’s digestibility. To ascertain the effects of endogenous protein and lipid on the starch digestibility of triticale flour, a triticale flour without proteins and/or lipids was studied. As shown in Figure 7, for both the raw and CHP samples, removing proteins and/or lipids from the triticale flour led to higher RDS and lower RS contents. Similarly, Ye et al. [18] found that after the removal of endogenous proteins and/or lipids, the starch digestibility of rice was significantly improved. López-Barón et al. [58] also reached a consistent conclusion by adding hydrolyzed exogenous proteins to wheat starch. This may be related to three factors: first, proteins and lipids attach to the surface of starch granules, blocking the interaction of starch with digestive enzymes; second, proteins and lipids restrict the expansion of starch particles, thus reducing the surface area of starch particles; and third, amylose and lipids form a complex, which reduces the digestion of starch. Therefore, the high protein and lipid contents of the cereal variety, as well as adding exogenous proteins during processing, may be considered for decreasing the starch digestibility of cooked triticale.

## 4. Conclusions

To promote the application of triticale in a healthy daily diet, this study evaluated the effects of two popular domestic cooking methods on the phenolic content, phenolic acid composition, antioxidant activity, and starch digestibility of the triticale cultivar. As expected, cooking led to significant changes in the compositions of the starch and bioactive compounds, the antioxidant abilities, and the physical properties of the triticale. Moreover, the two cooking methods led to significant differences in some characteristics of the triticale. Compared with the CAP sample, the CHP counterpart had a high RS content and a low GI, as well as high total phenolic and total phenolic acid contents. In addition, a difference in the microstructure and thermodynamic properties between the CAP and CHP samples was also found. According to the contents of the bioactive compounds, antioxidant activities, and starch digestibility, CHP may be a more appropriate cooking method for triticale porridge than CAP. These results may give some instructions for the domestic cooking of triticale as a health-promoting staple food. Future work may be done on screening triticale varieties with low starch digestibility and adding exogenous proteins during triticale processing to decrease starch digestibility and improve protein quality.

## Figures and Tables

**Figure 1 foods-13-00230-f001:**
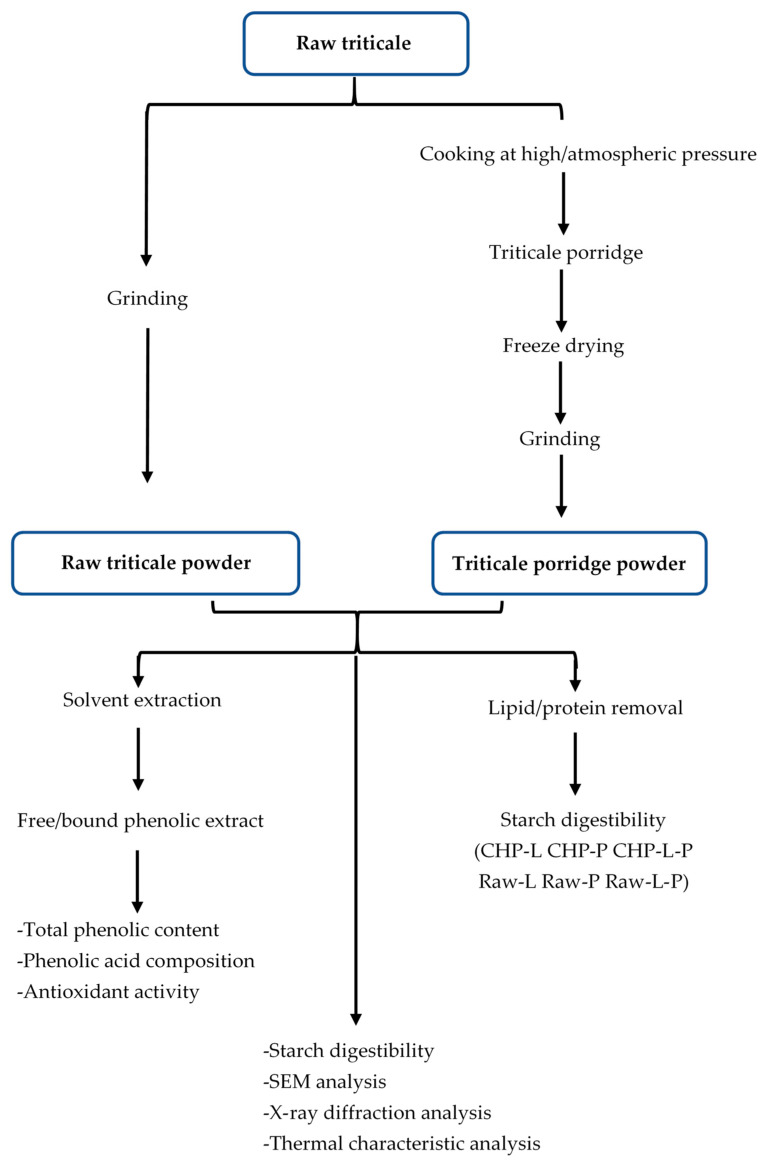
Diagram of sample preparation and analysis. Raw-L, Raw-P, and Raw-L-P represent uncooked triticale with the removal of lipids, proteins, and both lipids and proteins, respectively. CHP-L, CHP-P, and CHP-L-P represent high-pressure-cooked triticale with the removal of lipids, proteins, and both lipids and proteins, respectively.

**Figure 2 foods-13-00230-f002:**
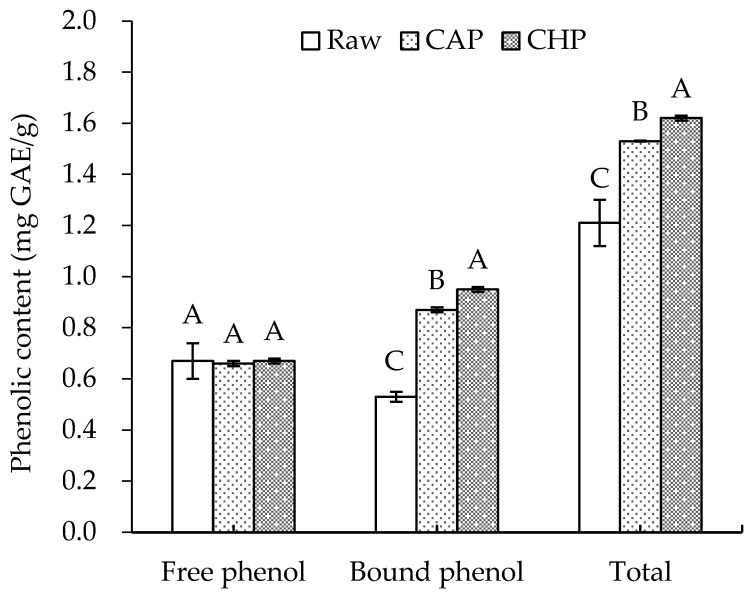
Free, bound, and total phenolic contents of raw and cooked triticale samples. CAP: cooked at atmospheric pressure. CHP: cooked at high pressure. The column and error bar represent mean and standard deviation (*n* = 3), respectively. Values with different capital letter are considered significantly different (*p* < 0.05).

**Figure 3 foods-13-00230-f003:**
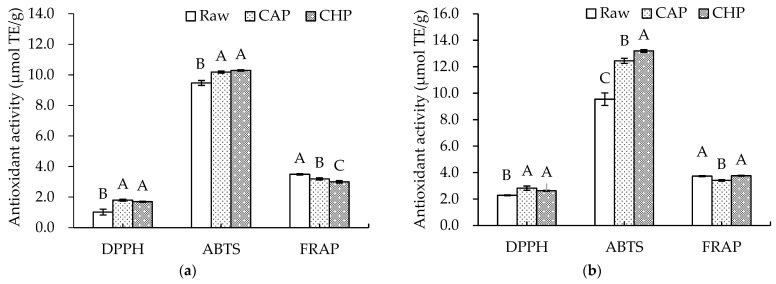
Antioxidant activities of phenolic extracts from raw and cooked triticale samples: (**a**) free phenolics; (**b**) bound phenolics. CAP: cooked at atmospheric pressure. CHP: cooked at high pressure. Columns and error bars represent mean and standard deviation (*n* = 3), respectively. Values with different capital letters are considered significantly different (*p* < 0.05).

**Figure 4 foods-13-00230-f004:**
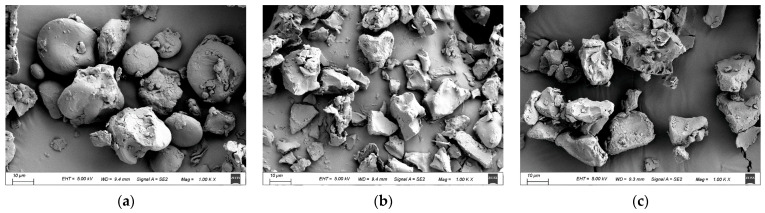
SEM micrographs of triticale samples: (**a**) raw; (**b**) CAP; (**c**) CHP. CAP: cooked at atmospheric pressure. CHP: cooked at high pressure.

**Figure 5 foods-13-00230-f005:**
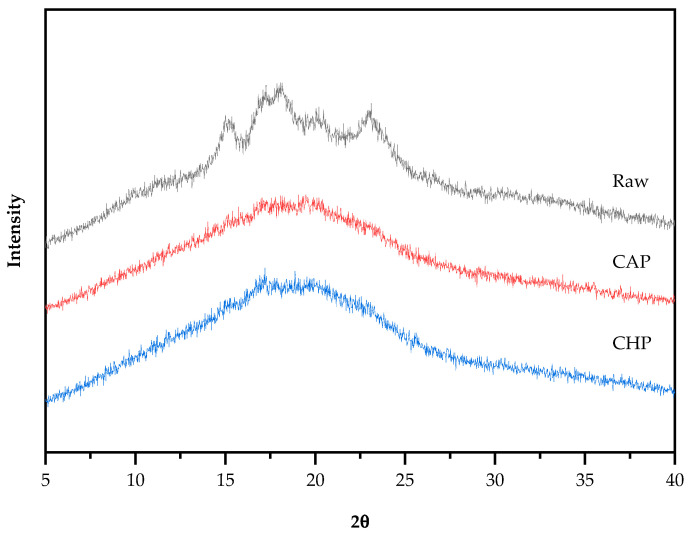
XRD spectra of raw and cooked triticale samples. CAP: cooked at atmospheric pressure. CHP: cooked at high pressure.

**Figure 6 foods-13-00230-f006:**
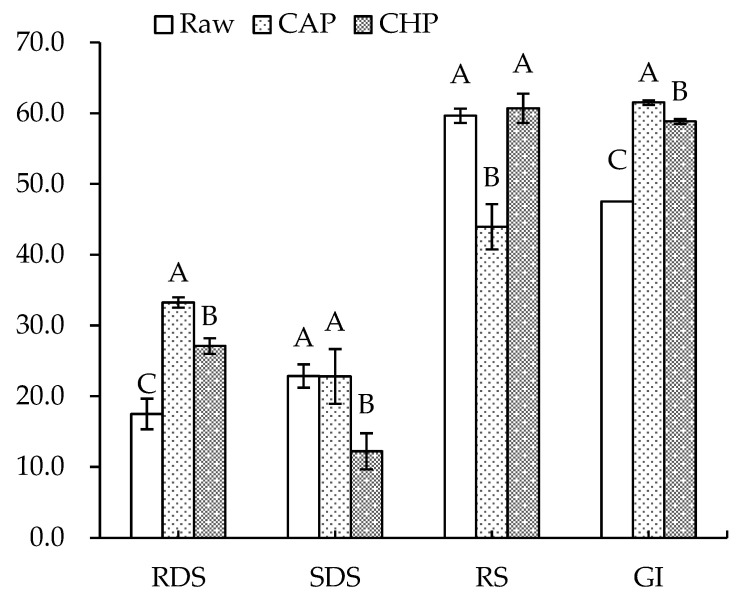
Contents of rapidly digestible starch (RDS), slowly digestible starch (SDS), resistant starch (RS), and glycemic index (GI) of raw and cooked triticale samples. CAP: cooked at atmospheric pressure. CHP: cooked at high pressure. Columns and error bars represent mean and standard deviation (*n* = 3), respectively. Values with different capital letters are considered significantly different (*p* < 0.05).

**Figure 7 foods-13-00230-f007:**
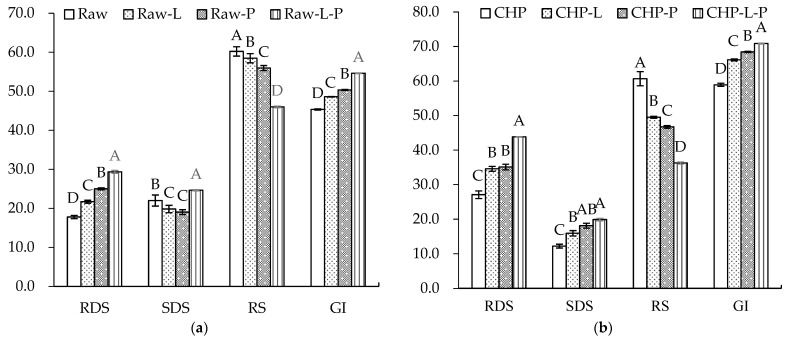
Contents of rapidly digestible starch (RDS), slowly digestible starch (SDS), resistant starch (RS), and glycemic index (GI) of triticale samples with the removal of lipids or/and proteins: (**a**) raw. (**b**) CHP, raw, uncooked triticale; CHP, high-pressure-cooked triticale; Raw-L, Raw-P, and Raw-L-P represent uncooked triticale with the removal of lipids, proteins, and both lipids and proteins, respectively. CHP-L, CHP-P, and CHP-L-P represent high-pressure-cooked triticale with the removal of lipids, proteins, and both lipids and proteins, respectively. Columns and error bars represent mean and standard deviation (*n* = 3), respectively. Values with different capital letters are considered significantly different (*p* < 0.05).

**Table 1 foods-13-00230-t001:** Compositions of free and bound phenolic acids in raw and cooked triticale samples (μg/g).

Compound	Treatment	Free	Bound
Protocatechuic acid	Raw	3.33 ± 0.21 ^C^	5.65 ± 0.69 ^A^
	CAP	7.57 ± 0.40 ^A^	2.80 ± 0.24 ^C^
	CHP	6.08 ± 0.10 ^B^	3.86 ± 0.27 ^B^
*p*-hydroxybenzoic acid	Raw	1.90 ± 0.07 ^B^	2.70 ± 0.35 ^A^
	CAP	2.42 ± 0.18 ^A^	1.23 ± 0.10 ^C^
	CHP	2.63 ± 0.08 ^A^	1.86 ± 0.09 ^B^
Vanillic acid	Raw	5.02 ± 0.50 ^C^	19.41 ± 1.93 ^A^
	CAP	7.85 ± 0.28 ^B^	10.32 ± 0.58 ^C^
	CHP	9.10 ± 0.24 ^A^	12.83 ± 0.19 ^B^
*t*-Cinnamic acid	Raw	18.16 ± 0.16	—
	CAP	—	0.28 ± 0.02 ^B^
	CHP	—	0.39 ± 0.02 ^A^
Gallic acid	Raw	—	—
	CAP	—	—
	CHP	14.34 ± 0.72	—
Syringic acid	Raw	1.99 ± 0.21 ^B^	5.95 ± 0.11 ^A^
	CAP	3.17 ± 0.41 ^A^	4.11 ± 0.31 ^B^
	CHP	2.14 ± 0.17 ^B^	4.67 ± 0.42 ^B^
Caffeic acid	Raw	1.36 ± 0.18 ^A^	8.43 ± 0.11
	CAP	—	—
	CHP	0.15 ± 0.02 ^B^	—
*p*-Coumaric acid	Raw	3.75 ± 0.09 ^A^	15.94 ± 0.32 ^A^
	CAP	0.59 ± 0.05 ^B^	—
	CHP	0.59 ± 0.04 ^B^	3.21 ± 0.19 ^B^
Ferulic acid	Raw	6.85 ± 0.03 ^B^	782.49 ± 1.30 ^A^
	CAP	10.46 ± 1.03 ^A^	247.82 ± 4.83 ^C^
	CHP	9.72 ± 0.21 ^A^	409.26 ± 3.16 ^B^
Sinapic acid	Raw	0.62 ± 0.05 ^C^	27.54 ± 0.49 ^A^
	CAP	4.81 ± 0.42 ^A^	17.96 ± 0.40 ^B^
	CHP	4.05 ± 0.19 ^B^	18.04 ± 1.05 ^B^
Total	Raw	42.96 ± 1.18 ^B^	868.11 ± 4.86 ^A^
	CAP	36.88 ± 2.10 ^C^	284.52 ± 4.56 ^C^
	CHP	48.79 ± 1.07 ^A^	454.12 ± 4.42 ^B^

—: not detected. CAP: cooked at atmospheric pressure. CHP: cooked at high pressure. For each individual phenolic acid, values with a different capital letter in the same column are considered as significantly different (*p* < 0.05).

**Table 2 foods-13-00230-t002:** Relative crystallinity and thermal characteristics of raw and cooked triticale samples.

Treatment	Relative Crystallinity	Thermal Characteristics				
		*T*o (°C)	*T*p (°C)	*T*c (°C)	*T*c–*T*o (°C)	∆*H* (J/g)
Raw	11.47 ± 0.15 ^A^	65.21 ± 0.12 ^A^	71.10 ± 0.87 ^A^	70.11 ± 0.12 ^A^	4.89 ± 0.09 ^C^	5.61 ± 0.03 ^A^
CAP	4.97 ± 0.23 ^B^	48.14 ± 0.09 ^C^	51.45 ± 0.10 ^B^	58.94 ± 0.05 ^B^	10.80 ± 0.11 ^A^	1.86 ± 0.14 ^C^
CHP	5.27 ± 0.06 ^B^	48.92 ± 0.59 ^B^	51.46 ± 0.09 ^B^	56.63 ± 0.14 ^C^	7.71 ± 0.62 ^B^	2.22 ± 0.11 ^B^

*T*o: onset temperature. *T*p: peak temperature. *T*c: conclusion temperature. ∆*H*: gelatinization enthalpy. CAP: cooked at atmospheric pressure. CHP: cooked at high pressure. Values with different capital letters in the same column are considered significantly different (*p* < 0.05).

## Data Availability

Data are contained within the article.

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
