# Peer review of "Impact of Cooking Methods on Phenolic Acid Composition, Antioxidant Activity, and Starch Digestibility of Chinese Triticale Porridges: A Comparative Study between Atmospheric Pressure and High Pressure Boiling"

_foods, 2024, doi:10.3390/foods13020230_

Round 1
Reviewer 1 Report
Comments and Suggestions for Authors
The article is very well written, it is very interesting both from a scientific and technological point of view.
I suggest indicating the amount of raw material in kilograms used in this experiment and it is also suggested to insert a figure that helps the reader visualize how the experimental part was carried out step by step.
Author Response
Thank you for your kind advice. Please see the attachment.

Reviewer 2 Report
Comments and Suggestions for Authors
Manuscript ID: foods-2798771
Title: Impact of cooking methods on phenolic acid composition, anti-2 oxidant activity, and starch digestibility of Chinese triticale 3 porridges: a comparative study between atmospheric pressure 4 and high pressure boiling
Abstract
The abstract is specific and concise, but aim of the study should be reformulated for better understanding for readers. Moreover, sentence ‘ Our findings suggested that CHP may be a more suitable cooking method for lower-GI triticale porridge‘ does not refer to the studies carried out, and is a very far-reaching conclusion - it has no basis in the results obtained or the analyses carried out. It should be removed.
Introduction
The introduction is appropriate to the topic.
Materials and Methods
The ‘Material & Methods’ is correct. Information about, How many repetitions were made in every assay? should be added.
2.3. Short information about this methods should be added here, not only mentioned the reference.
2.6. GA – explain abbreviation in bracket
2.7. Nowadays, DAD detector could be followed by another eg. MS, to get more precise results. In my opinion authors should mention about it (mayve they plan it in their future studies…) in section Results or Discussion.
‘In vitro’ always in italic through the text – should be improved.
Results
The results were discussed adequately and sufficiently.
The ‘Conclusions’ are correct.
Author Response

(The authors gave the same response as above.)
